# Acceleration, simplification and potential parallelization of digital polymers sequencing by coupling tandem mass spectrometry with ion mobility

Isaure Sergent[1], Georgette Obeid [2], Thibault Schutz[2], Jean-François Lutz [2] ✉ & Laurence Charles [1] ✉

Tailoring the structure of digital polymers is an efficient strategy for reliable reading of large amounts of data by tandem mass spectrometry. Notably, full sequence coverage of chains containing up to 33 bytes of information is achieved for block-truncated poly(phosphodiester)s designed to undergo controlled fragmentations. However, the previously established reading methodology based on multiple MS stages performed sequentially remains slow and not prone to automation. Here, we report a full gas-phase bottom-up workflow enabling production, separation and sequencing of all subsequences of block-truncated poly(phosphodiester)s in a single run. To do so, a multidimensional coupling involving two activation stages in tandem with ion mobility spectrometry has been optimized. Since blocks to be sequenced have their mobility varying in a predictable manner, proper selection of tags used for their identification permits to achieve mobility resolution prior to sequencing. Performing this coupling with MALDI further paves the way to automated imaging-based reading approaches.

Bottom-up approaches are efficient strategies implemented to overcome limitations of tandem mass spectrometry (MS/MS) for de novo sequencing of (bio)polymers, for which full sequence coverage is barely achieved for chains with more than 50 repeating units when using collision-induced dissociation (CID). In bottom-up workflows, long chains are sub-divided into smaller pieces that can be sequenced individually, and the original primary structure is reconstructed by proper reassembly of these sub-sequences[1]. For biopolymers such as proteins[2] or oligonucleotides[3], enzymes are employed to perform chain cleavages at specific locations, yielding small segments amenable to MS/MS sequencing while looking for regions of overlap enables reassembling the primary structure. In both cases, enzymatic digestion is part of sample preparation and so-obtained mixtures

are subjected to chromatographic separation for best ionization of each component and, most importantly, safe selection of precursor ions to be fragmented. The same strategy has been applied to read information stored in digital polymers in which monomers are defined as molecular bits[4], yet in a less time-consuming, full gas-phase workflow. Indeed, the structure of these synthetic macromolecules can be specifically designed to be cleaved upon collision activation and produce non-isobaric subsequences in the gas phase. This was demonstrated for sequence-defined poly(phosphodiester)s (PPDEs) in which alkoxyamine groups are placed between each set of eight bits (i.e. one byte) and each byte is labeled by a mass tag[5]. Using soft CID conditions in MS[2] allows selective cleavage of the weakest NO−C bonds, yielding a series of mass tag-shifted intact blocks that can further

[1]Aix Marseille Université, CNRS, Institut de Chimie Radicalaire (ICR), Marseille, France. [2]Université de Strasbourg, CNRS, Institut de Science et d'Ingénierie Supramoléculaires (ISIS), Strasbourg, France. ✉e-mail: jflutz@unistra.fr; laurence.charles@univ-amu.fr

be individually activated for sequencing in pseudo-MS[3] experiments (Fig. 1a). The MS[2] spectrum also contains fragments composed of multiple blocks that are used as regions of overlap to reconstruct the whole polymer sequence. Further improvements of the molecular structure of these block-truncated PPDEs (b-PPDEs)[6] have enabled high-capacity digital polymers containing up to 33 bytes per chains to be efficiently sequenced, with no error, using a routine mass spectrometer[7]. The overall analytical workflow developed to retrieve binary information stored in b-PPDEs is highly reliable but it lacks efficiency as it involves multiple successive experiments, notably a number of MS[3] sequencing events (3 min each) which increases with the number of bytes in the chain.

Here, we report that the whole readout process can be advantageously accelerated by performing serial (instead of sequential) sequencing of blocks by combining the two activation steps with ion mobility spectrometry (IMS) in a single MS-(CID)-IMS-(CID)-MS experiment (Fig. 1b). Taking advantage of the instrumental configuration of the Synapt mass spectrometer, intact b-PPDE chains are mass selected as primary precursor ions in the Q1 quadrupole, activated in the ion trap placed in front of the traveling wave ion mobility (TWIM) cell so that individual blocks released as primary fragments can reach the post-IMS collision cell one after the other for individual sequencing, using secondary fragments measured by the orthogonal acceleration time of flight (oa-TOF) mass analyzer. Performance of this coupling for serial sequencing of b-PPDE blocks relies on proper selection of the mass tags, originally conceived to prevent mass coincidence of blocks in MS[2] and also employed here as "shape tags" to ensure gas-phase separation of blocks in the IMS cell.

## Results and discussion

### IMS behavior of PDE blocks

IMS is a gas-phase separation method allowing spatial dispersion of ions according to their mobility across a cell filled with a buffer gas under the influence of an electric field[8]. The arrival time ($t_A$) of an ion while exiting the mobility cell depends on its charge state and collision cross section (CCS), which represents its apparent surface area as determined by its three-dimensional shape. In Synapt instruments, the TWIMS technology employs a series of potential waves to propel ions through the buffer gas inside the mobility cell[9]. Mobility of ions with larger CCS is reduced below the wave velocity since they experience more collisions with the gas: this makes them roll over the potential wave crest more often than do ions with lower CCS that hence travel faster. Optimization of ion separation in TWIMS is typically performed by varying two main parameters, the wave velocity (WV) and the wave height (WH). To investigate the capability of this IMS technique at resolving individual PDE blocks, MS-(CID)-IMS-MS experiments were performed for a set of PPDE triblocks available from a previous study (Supplementary Table 1)[10]. In these experiments, the entire chain is selected as the precursor ion (first MS step), activated via collisions with argon (CID step) to release blocks that are mass measured as intact species (second MS step) after they have gone through the IMS cell. In negative mode electrospray ionization (ESI), b-PPDEs are produced as gas-phase anions distributed over different charge states: selecting abundant precursor ions at a charge state equal to a multiple of the number of blocks is key to minimize signal dilution and hence guarantee detectability of sequencing fragments in MS[3] (Supplementary Fig. 1). Accordingly, all polymers submitted to CID were selected with two charges per block. Owing to the design of b-PPDEs, individual Bi blocks exhibit different structures as a function of their original location in the chain (Fig. 2a). These blocks can be described as small

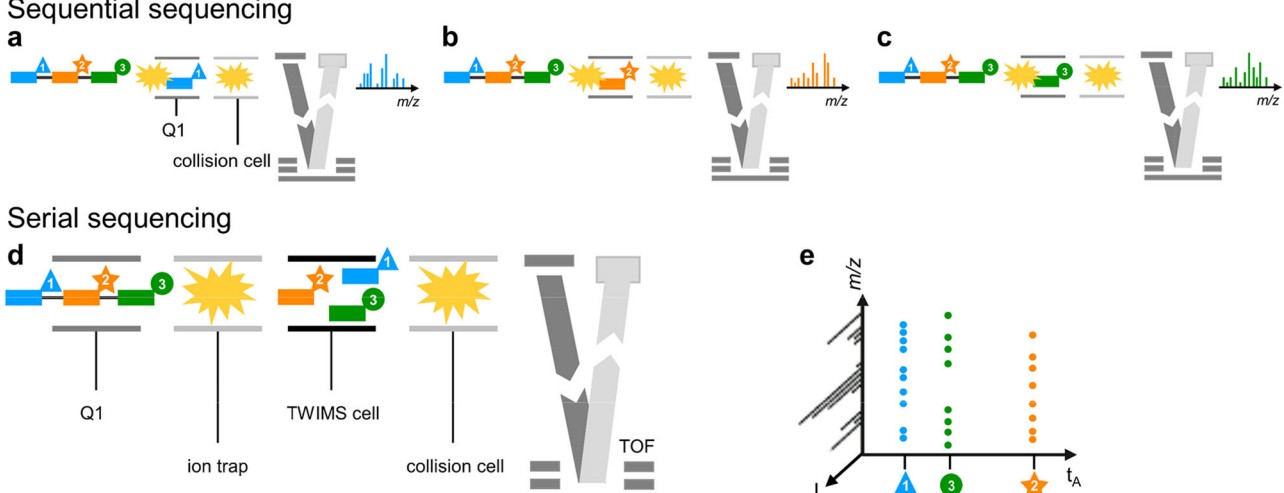

**Fig. 1 | Instrumental configurations for sequencing of b-PPDEs illustrated with triblock chains.** Top: The previously established reading methodology performing block sequencing in a sequential manner employs an ESI-Q-TOF mass spectrometer with a single collision cell. Intact b-PPDE chains are thus first activated (yellow explosion symbol) at the entrance of the mass spectrometer (in-source activation) to release blocks that further need to be individually mass selected in the Q1 quadrupole in different CID experiments, as shown for **a** the first block (in blue), **b** the second block (in orange) and **c** the third block (in green) of the b-PPDE triblock. Tags labeling each block (blue triangle, orange star, green circle) avoid their mass coincidence. After selection, each block enters the collision cell for activation and fragments are measured in the TOF mass analyzer. Three individual CID spectra are obtained, each recorded for 3 min. Bottom: The MS-CID-IMS-CID-MS reading methodology proposed in the present study makes use of the unique configuration of Synapt mass spectrometers, with an IMS cell placed between two activation devices, to perform serial block sequencing. **d** Intact b-PPDE chains are introduced in the instrument where they are selected in the Q1 quadrupole based on their m/z value (first MS step). These primary precursors are then directed into the ion trap where they are activated to break down into their constituting blocks (first CID step). These primary fragments are injected into the IMS cell where they get separated based on their CCS (IMS step): here, tags are employed to mass-shift the blocks and to assist their IMS resolution. Once separated, blocks can reach the collision cell one after the other to be activated individually (second CID step). Secondary fragments generated from each block are then analyzed by the TOF mass analyzer (second MS step). **e** The output of this coupling (acquisition time: 5 min, regardless of the number of blocks) is a 3D plot displaying the intensity (I) of sequencing fragments as a function of their m/z value, all aligned at the arrival time ($t_A$) of their respective block precursor in IMS. Adapted with permission from ref. 12. Copyright 2024 American Chemical Society.

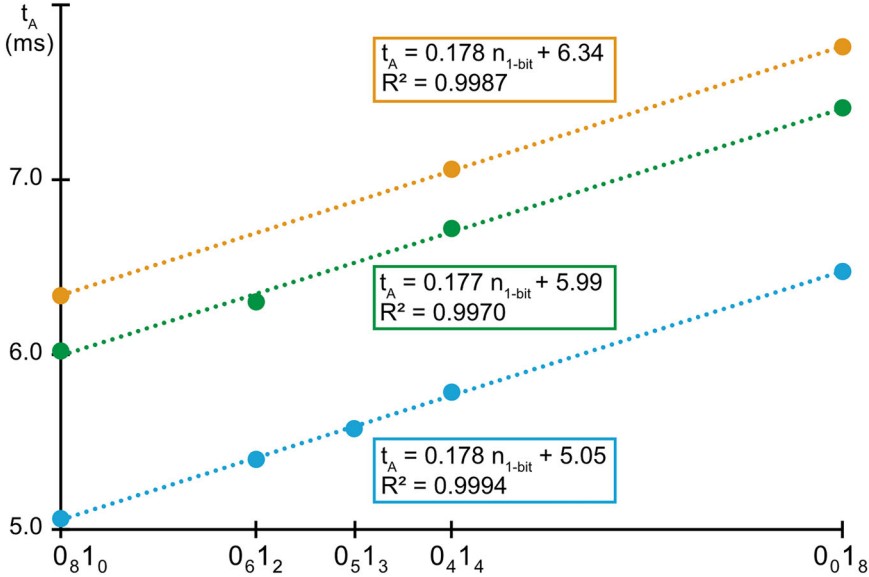

**Fig. 2 | Structural features in digital b-PPDEs. a** General structure of intact chains. **b** Structure of individual blocks released upon homolysis of alkoxyamine bonds in b-PPDE as a function of their initial location: first (blue), inner (orange) and last (green). Coding units are defined as 0-bits when R = H (138.0 Da) or 1-bits when R = CH₃ (166.1 Da). Ti (in black) designates mass tag selected amongst moieties shown in (**c**).

**Fig. 3 | Variation of $t_A$ (in ms) of blocks as a function of their $0_x1_y$ comonomeric composition.** Measuring block $t_A$ as their number of 1-bits increases permits to establish linear relationships with the same slope, regardless of the block category, as shown for first block (in blue), inner block tagged with T2 (in orange) and last block tagged with T8 (in green).

oligomers containing one string of bits and different end-groups: as depicted in Fig. 2b, the first block (in blue) is a PDE octamer still holding the original HO α-group and a carbon-centered radical moiety as ω, inner blocks (in orange) are PDE octamers with a nitroxide holding one tag (Ti, Fig. 2c) as α and a carbon-centered radical as ω, and the last block (in green) is a PDE heptamer with one tag attached to its nitroxide α-group and an ω-group containing the last alkyl coding segment attached to the original OH termination of the chain. These structural differences have two analytically useful consequences. On the one hand, the first block is always the lightest one, regardless of its sequence, and can thus be readily identified based on its $m/z$ value in MS/MS (Supplementary Fig. 1b). On the other hand, $t_A$ measured for [Bi − 2H]²⁻ ions in IMS are observed to depend on the block category (first,

inner or last), as well as on their 0/1 co-monomeric composition and the structure of their tag but, interestingly, not on their sequence (Supplementary Fig. 2). More precisely, for blocks with the same "category/tag" combination, measured $t_A$ values are observed to increase linearly with the number of 1-bits in their sequence (Fig. 3). In the optimized experimental conditions used to record these data (Supplementary Fig. 3), changing any one unit from 0- to 1-bit always leads to a $t_A$ increase of 0.18 ms, as measured by the slope of the linear trends. Using appropriate procedure to calibrate the TWIM cell[11], we recently showed that this $t_A$ increase corresponds to a + 5 Å² increment of the block CCS. Molecular modeling has evidenced that the coded segment arranges into a random coil which grows as one propyl segment coding for 0-bit is changed to a bulkier 2,2-dimethyl-propyl

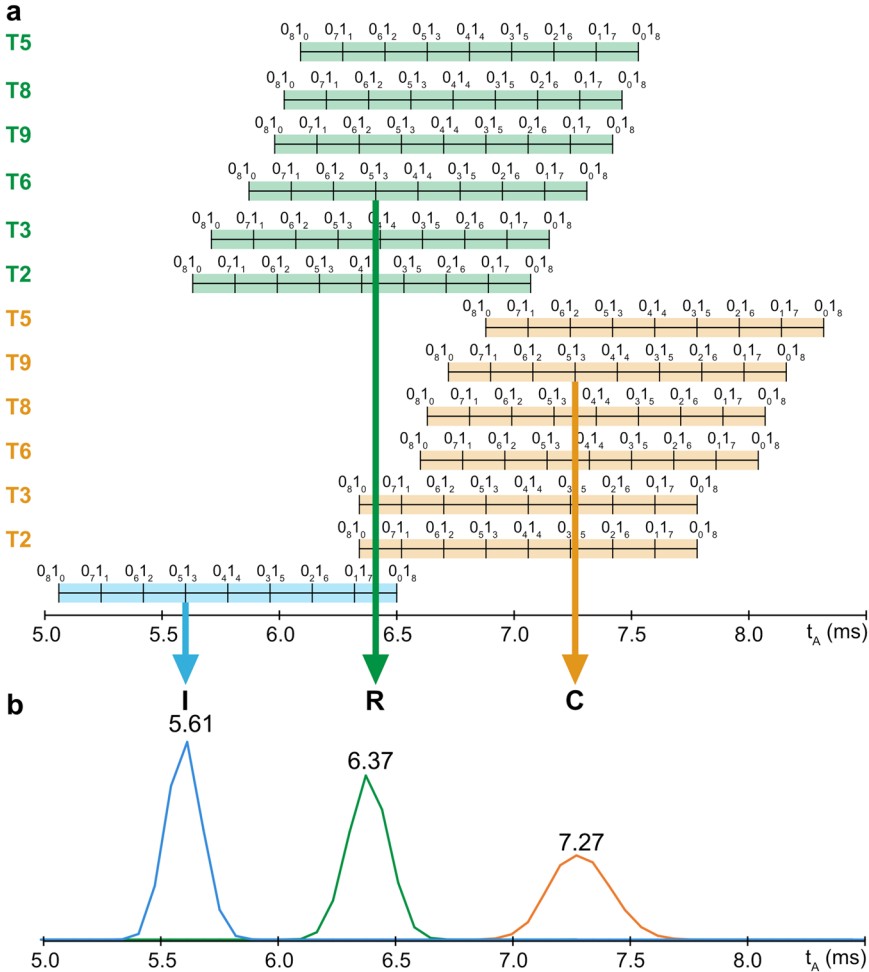

**Fig. 4 | Selection of tags for the PPDE triblock coding for "ICR". a** Predicted $t_A$ ranges (WV = 30 V, WH = 550 m.s$^{-1}$) for doubly deprotonated blocks as a function of their category (first: blue, inner: orange, last: green) and their Ti tag indicated on the left. This chart was used to design the PPDE triblock coding for "ICR" with all blocks of the same $0_5 1_3$ composition: based on $t_A$ = 5.58 ms expected for the first block (B1) coding for "I", tag T6 was selected to label the last block (B3) coding for "R" while tag T9 was selected to label the inner block (B2) coding for "C". **b** IMS traces extracted for [B1 – 2H]$^{2-}$ at $m/z$ 667.6 (in blue), [B2 – 2H]$^{2-}$ at $m/z$ 886.2 (in orange) and [B3 – 2H]$^{2-}$ at $m/z$ 777.7 (in green) after performing MS-(CID)-IMS-MS (selected precursor ion: $m/z$ 777.2; collision energy: 26 eV, laboratory frame) of the PPDE triblock coding for "ICR".

segment coding for 1-bit[12]. This peculiar gas-phase conformation also explains why the 0/1 sequence has not influence on the block CCS. Overall, data displayed in Fig. 3 suggest that experimental measurement of one block is sufficient to predict $t_A$ of any other blocks with the same "category/tag" combination but different co-monomeric compositions, which would advantageously save time and synthesis efforts.

Another interesting feature displayed in Fig. 3 is that all lines have different y-intercepts that vary as a function of the Ti tag decorating either inner or final blocks (*vide infra*), hence indicating that the tag also influences the IMS behavior of blocks. This was confirmed by molecular modeling of inner blocks showing that, in the tridimensional conformation adopted by these doubly charged species, the tag points out of the coil formed by the coded segment[12]. However, the CCS influence of tags was observed to change with the block category but no clear trend could be identified. Moreover, values measured for these y-intercepts are too close to ensure IMS separation of all blocks regardless of their 0/1 composition. For example, it can be anticipated from Fig. 3 that, if they were produced from the same b-PPDE precursor, the $0_6 1_2$ final block labeled with T8 ($t_A$ = 6.37 ms, in green) would not be resolved from the $0_8 1_0$ inner block labeled with T2 (in orange) experimentally measured at $t_A$ = 6.34 ms. Nevertheless, knowing the binary message to be written in a digital b-PPDE (and so

the 0/1 composition of its blocks) should permit to define, in advance of synthesis, which tags need be used to ensure IMS separation of all blocks.

**Predictive distribution of tags in b-PPDEs**

Based on the assumption that the IMS behavior of [Bi – 2H]$^{2-}$ species is predictable, data recorded in MS-(CID)-IMS-MS experiments were used to define $t_A$ range for all blocks as a function of their "category/tag" combination (Supplementary Table 2). This is documented in Fig. 4a which displays the $t_A$ range expected for the first block (in blue) as compared to those predicted for inner (in orange) and final (in green) blocks as a function of their Ti tag indicated on the left. Such a chart permits to anticipate, at a glance, which tags need to be selected to ensure IMS separation of all blocks.

Robustness of the proposed predictive approach was first evaluated with a triblock b-PPDE coding for the "ICR" acronym of our laboratory. Using the ASCII code, the binary sequence for these letters is such as 01001001 for I, 01000011 for C and 01010010 for R, all corresponding to the same $0_5 1_3$ co-monomeric composition. According to Fig. 4a, the first block (B1, in blue) is expected at $t_A$ = 5.58 ms: accordingly, selecting tag T6 to label the last block (B3, in green, expected at $t_A$ = 6.41 ms) and tag T9 to label the inner block (B2, in

orange, expected at $t_A = 7.23$ ms) should ensure optimal IMS separation of the three blocks.

The corresponding b-PPDE (Supplementary Fig. 4a) was then synthesized and first analyzed by MS-(CID)-IMS-MS after selecting this triblock polymer at the 6− charge state ($m/z$ 777.2) as the precursor ion. Individual IMS traces extracted for $[B1 - 2H]^{2-}$ at $m/z$ 667.6 (in blue), $[B2 - 2H]^{2-}$ at $m/z$ 886.2 (in orange) and $[B3 - 2H]^{2-}$ at $m/z$ 777.7 (in green) are superimposed in Fig. 4b: $t_A$ experimentally measured for each block (with ±0.07 ms precision) perfectly match with predicted values. Clear baseline resolution is achieved for the three blocks, indicating that they will reach the post-IMS collision cell independently of each other as required to ensure the "purity" of their respective CID spectra. The $m/z$ 777.2 precursor ion was then subjected to MS-(CID)-IMS-(CID)-MS. Using the same 34 eV collision energy (laboratory frame) for all $[Bi - 2H]^{2-}$ ions in the second activation stage still allows these secondary precursors to be detected in their respective CID spectrum while generating nearly all members of the eight fragment series expected from cleavage of all phosphate bonds in PDE monomers (Supplementary Fig. 5). As also supported by accurate mass measurements (Supplementary Tables 3–5), full sequence coverage of all blocks is achieved from these CID data. Most importantly, the acquisition time for this experiment was 5 min, that is, nearly two-fold less time than the 9 min (3 min per block) typically requested for sequencing triblock b-PPDEs with the previous sequential method.

Due to the low resolving power of the TWIMS device, IMS peaks displayed in Fig. 4b are quite large and, because of diffusion effects in the IMS cell, peak width increases with $t_A$ (half height peak width of 0.20 ms for B1 at $t_A = 5.61$ ms, 0.23 ms for B3 at $t_A = 6.37$ ms and 0.36 ms for B2 at $t_A = 7.27$ ms). This is expected to raise issues when increasing the number of blocks to be resolved in the proposed serial sequencing experiment. Actually, the longest b-PPDE amenable to this coupling was a pentablock, specifically designed for the sake of demonstration. Indeed, to maximize the probability for the five blocks to be separated in IMS, the coded message had to be composed with characters spanning a wide range of 0/1 co-monomeric compositions. Typically, the first and the last characters must have binary sequence rich in 0-bits to ensure early IMS elution of their respective blocks so that the largest $t_A$ range is available for inner blocks, the co-monomeric composition of which also needs to be clearly distinct. To fulfill these criteria, we considered the ASCII code defining letters of the Latin alphabet (Supplementary Table 6) to arrange a series of different characters including uppercase and lowercase letters as well as punctuation mark, not necessarily aiming at a meaningful five-letter word. The so-obtained coded message is "PHew!", performing tag selection as shown in Fig. 5a. Based on $t_A = 5.40$ ms expected for the $0_6 1_2$ B1 coding for "P" (01010000), it was decided to use one punctuation mark which sequence contains many 0-bits in the ASCII code for the last character: we chose "!" (01000001) and tag T2 to label this $0_6 1_2$ last B5 block so that it is expected to elute at $t_A = 5.98$ ms. Then, three characters with very distinct 0/1 compositions were selected for inner blocks in order to maximize their IMS separation after proper tag selection. Accordingly, tag T2 was chosen to label the $0_6 1_2$ B2 block coding for "H" (01001000) hence expected at $t_A = 6.71$ ms, as well as the $0_4 1_4$ B3 block coding for "e" (01100101), so expected at $t_A = 7.06$ ms, whereas the $0_2 1_6$ B4 block coding for "w" (01110111) is expected at the later $t_A = 7.76$ ms when labeled with tag T9. The corresponding b-PPDE (Supplementary Fig. 4b) was then synthesized and, after ionization in negative mode ESI, the MS-(CID)-IMS-MS experiment was performed, selecting the $m/z$ 807.4 precursor ion corresponding to the polymer at the 10− charge state. As illustrated by IMS traces extracted for all blocks (Supplementary Fig. 6), experimental $t_A$ values are in line with predicted values as they are measured at 5.40 ms for $[B1 - 2H]^{2-}$, 6.72 ms for $[B2 - 2H]^{2-}$, 7.06 ms for $[B3 - 2H]^{2-}$, 7.76 ms for $[B4 - 2H]^{2-}$ and 6.02 ms for $[B5 - 2H]^{2-}$. Baseline separation is obtained for all blocks but B2 and B3; however, the achieved resolution is still sufficient to envisage extraction of "pure" CID spectra. This has been confirmed when running the MS-(CID)-IMS-(CID)-MS, as shown in the 3D-plot of Fig. 5b which displays abundance of sequencing fragments as a function of their $m/z$ value and arrival time in IMS. This 3D-plot clearly shows that, as expected, all fragments align at the $t_A$ of their respective precursor. As mentioned before, IMS peaks corresponding to B2 and B3 are not fully resolved but they both exhibit a 0.30 ms $t_A$ range free of interference (Supplementary Fig. 6) to enable extraction of "pure" CID spectra, which was sufficient to obtain sequencing fragments of good signal-to-noise ratio (*vide infra*). In Fig. 5b, $[B1 - 2H]^{2-}$ is observed with a much lower abundance compared to all other secondary precursors. Due to the large number of 0-bits in the B1 sequence combined with the lack of tag for this first block, the $[B1 - 2H]^{2-}$ ion has a significantly smaller $m/z$ 653.6 value than other blocks ($[B5 - 2H]^{2-}$ at $m/z$ 738.7, $[B2 - 2H]^{2-}$ at $m/z$ 844.2, $[B3 - 2H]^{2-}$ at $m/z$ 872.2, and $[B4 - 2H]^{2-}$ at $m/z$ 928.3). Since all blocks were subjected to the same activation regime in the second CID step (34 eV, laboratory frame), the lightest B1 has fragmented to a higher extent compared to other blocks. Nevertheless, full sequence coverage of all blocks was achieved from these data, allowing error-free reconstruction of the original "PHew!" coded message as detailed in Supplementary Fig. 7 and Supplementary Tables 7–11. In addition, acceleration of the reading process is more substantial than in the case of the "ICR" triblock as it took 5 min to obtain CID data for all five blocks as compared to 15 min when performing sequencing of five blocks in a sequential manner.

It should be acknowledged that this five-byte coded message was specifically designed to demonstrate the capability of the MS-(CID)-IMS-(CID)-MS coupling for on-line sequencing of b-PPDE pentablock. Overall, practical use of this analytical workflow is limited to reading any three bytes of information with no restriction on bit composition (when implemented in our instrument). Performance of the coupling would be substantially improved when performed with next-generation TWIM technologies such as cyclic IMS offering resolving powers above 300[13], compared to the 30–40 resolution achieved in the 25 cm linear TWIM cell[14]. Taking advantage of another storage dimension can alternatively compensate for limited IMS resolution. This could typically be done by writing coded messages with multiple b-PPDE triblocks spatially organized on a substrate and reading them via an imaging approach with an ionization technique with surface sampling capabilities.

## Surface sampling in-line with serial sequencing

Preliminary works towards the proposed imaging approach were then performed, first evaluating desorption/ionization techniques available in our instrument for their capabilities to produce deprotonated b-PPDEs with high yield. A key requirement for this coupling to produce readable data is indeed production of highly abundant ions due to significant signal loss during the two activation steps and the IMS separation, in addition to dilution of precursor ion signal over primary and secondary fragments. The most appealing technique was desorption electrospray ionization (DESI)[15] because of the propensity of this method to generate multiply charged species as in ESI. DESI makes use of an electrospray probe to produce primary charged droplets of solvent to deposit a thin film on the surface to enable analyte dissolution. Impact of subsequent primary droplets onto this thin film gives rise to secondary droplets containing analytes, further emitted as intact ions in the gas phase according to an ESI process[16]. Generation of multiply charged species by DESI would thus permit to rely on the predictive IMS chart already established for doubly charged blocks. DESI has already been successfully used for imaging of digital oligomers[17,18] but, in the case of b-PPDEs, multideprotonated species were not produced with sufficient abundance to foresee detection of sequencing fragments using the MS-(CID)-IMS-(CID)-MS coupling. In contrast, high intensity signals were obtained using matrix-assisted

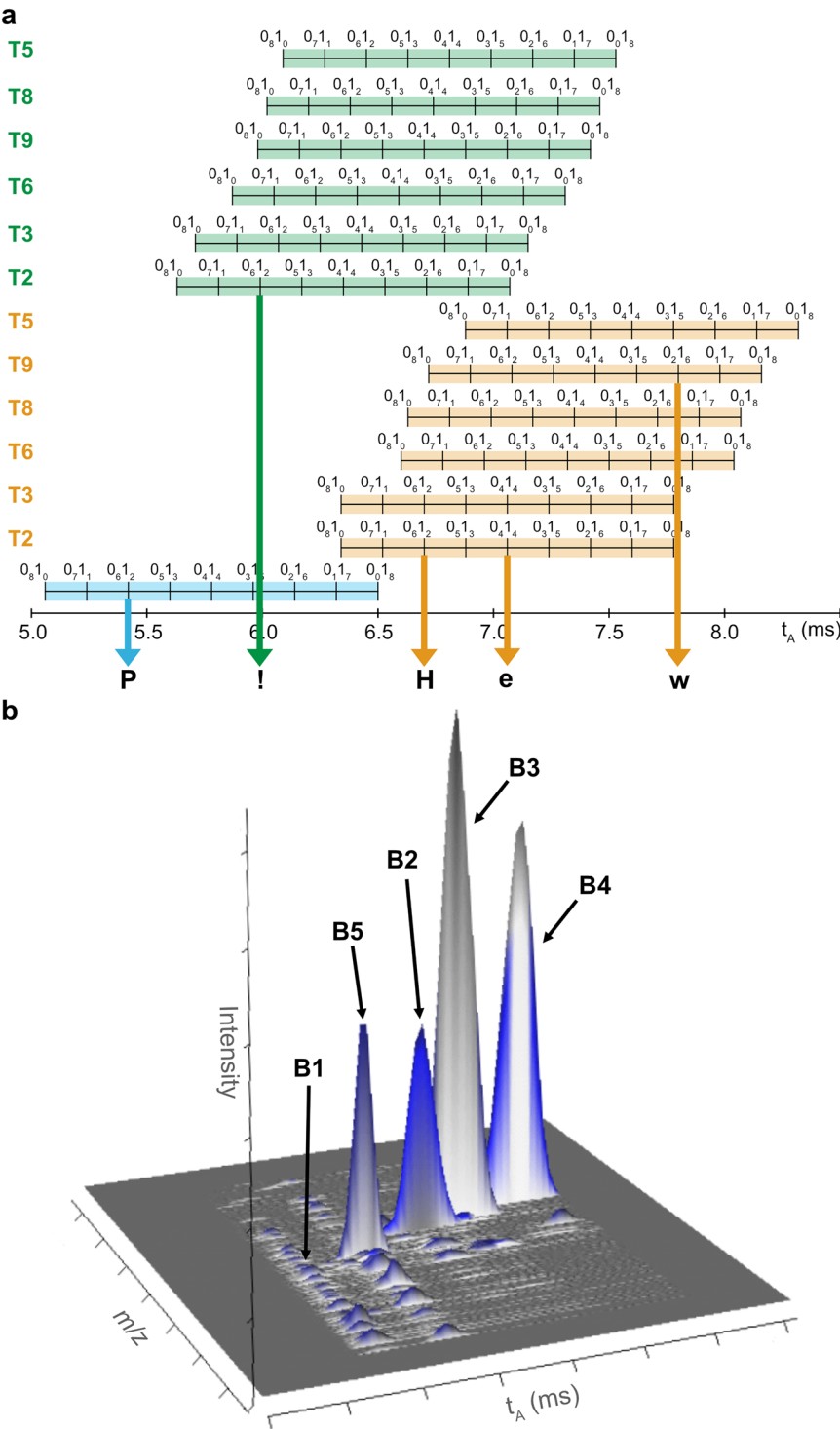

**Fig. 5 | Selection of tags for the PPDE pentablock coding for "PHew!".**
**a** Predicting chart of $t_A$ ranges (WV = 30 V, WH = 550 m.s$^{-1}$) for doubly deprotonated blocks as a function of their category (first: blue, inner: orange, last: green) and their Ti tag indicated on the left. This chart was used to design the PPDE pentablock coding for "PHew!": based on $t_A$ = 5.40 ms expected for B1 coding for "P" ($0_6 1_2$), tag

T2 was selected to label B5 coding for "!" ($0_6 1_2$); then, tag T2 was selected to label both B2 coding for "H" ($0_6 1_2$) and B3 coding for "e" ($0_4 1_4$) while tag T9 was selected to label B4 coding for "w" ($0_2 1_6$). **b** 3D-plot obtained from the MS-(CID)-IMS-(CID)-MS experiment performed for the PPDE pentablock coding for "PHew!".

laser desorption/ionization (MALDI) of triblock b-PPDEs prepared with α-cyano-4-hydroxycinnamic acid (CHCA) as the matrix. MALDI mass spectra do not exhibit any intact deprotonated b-PPDE chains as fragile C−ON bonds spontaneously cleave during the MALDI process[19]. Nevertheless, all useful primary fragments are clearly observed in MALDI-MS, as shown in Fig. 6a with the three individual blocks of the "ICR" polymer together with the two diblock fragments (B1B2 and

B2B3) to be employed to reconstruct the sequence of blocks. Occurrence of such selective dissociation in the MALDI source permits to simplify the coupling since only one CID stage is now required. Accordingly, the analytical workflow becomes MALDI-IMS-(CID)-MS, but introduction of all MALDI ions in the IMS cell with no prior mass selection raises two constraints. On the one hand, this requires b-PPDE samples to be free of any impurity, which is best achieved with the

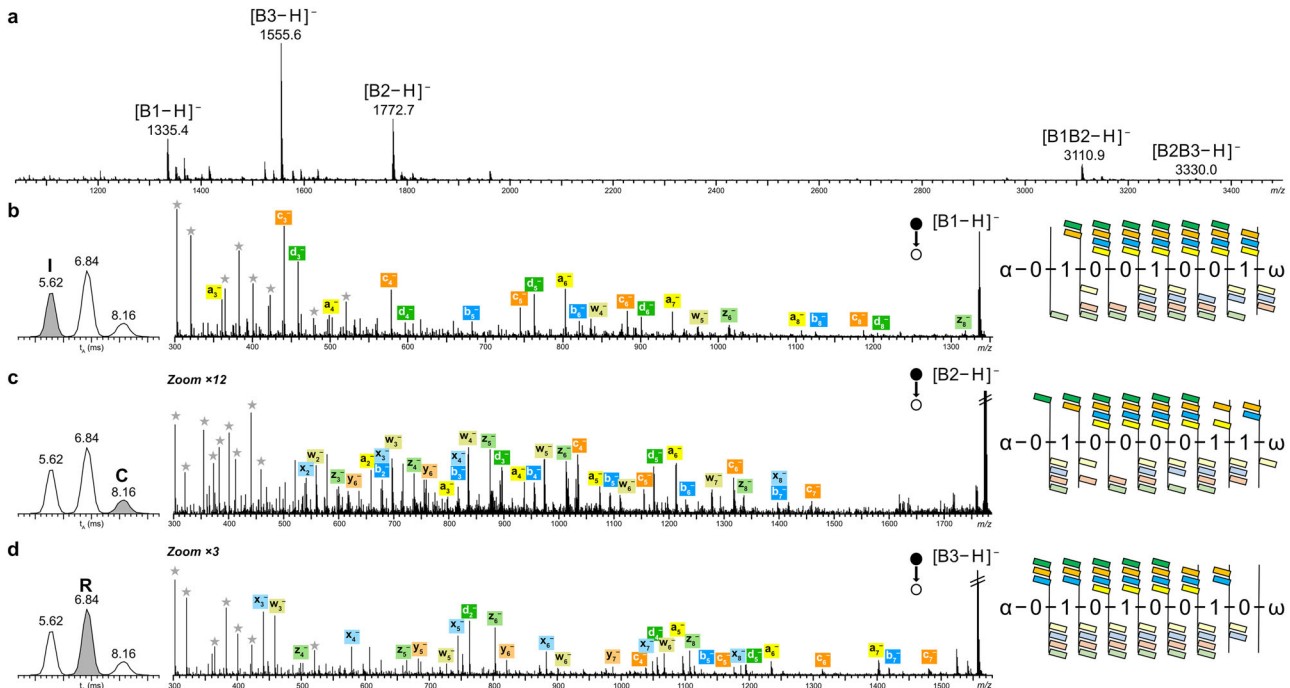

**Fig. 6 | Serial block sequencing after MALDI. a** MALDI-MS of the b-PPDE coding for "ICR". MALDI-IMS-(CID)-MS of the "ICR" polymer with superimposed extracted IMS traces obtained at WV = 25 V and WH = 130 m.s$^{-1}$ (left panel), corresponding dissociation spectrum (middle panel) and associated sequence coverage (right panel) for **b** [B1 – H]$^-$ at *m/z* 1335.4 and t$_A$ = 5.62 ms coding for I, **c** [B2 – H]$^-$ at *m/z* 1772.7 and t$_A$ = 8.16 ms coding for C, and **d** [B3 – H]$^-$ at *m/z* 1555.6 and t$_A$ = 6.84 ms coding for R. Gray stars are used to designate internal fragments in dissociation spectra.

implemented synthesis protocol when the number of blocks remains low. On the other hand, the TOF mass analyzer needs to be operated above *m/z* 300 to avoid sampling highly abundant ionic matrix clusters and prevent detector overloading. This means that sequencing fragments of low *m/z* will no longer be recorded, which should however not jeopardize full sequence coverage. The last question to be solved remains the capabilities of the TWIMS cell to efficiently separate singly charged blocks as produced by MALDI. After experimental conditions have been re-optimized using the same set of triblock oligomers (Supplementary Table 1), IMS resolution of singly deprotonated blocks was achieved with the same quality compared to their doubly charged congeners, as depicted by the new predictive chart established for [Bi – H]$^-$ ions (Supplementary Fig. 8). Based on this chart, the "ICR" polymer is perfectly suitable for the MALDI coupling since its three $0_5 1_3$ blocks have their t$_A$ expected at 5.62 ms for B1, 6.84 ms for B3 labeled with T6 and 8.10 ms for B2 labeled with T9. As shown in the IMS traces reported in the left panel of Fig. 6b–d, arrival times experimentally measured for deprotonated blocks are consistent with these predictive values. Sequencing schemes displayed in the right panel of Fig. 6b–d clearly show that full sequence coverage is achieved for all blocks despite fragments below *m/z* 300 are missing. This was the first time sequencing was performed for singly charged PPDE blocks and, in contrast to their doubly charged congeners, they require different activation energy regimes: while the lightest [B1 – H]$^-$ is best sequenced at 55 eV, [B2 – H]$^-$ and [B3 – H]$^-$ both require collision energy to be raised at 120 eV. Accordingly, the MALDI-IMS-(CID)-MS coupling was performed according to the MS$^E$ multiplex fragmentation mode[20], switching the CID energy from 55 to 120 eV along the whole run. An additional advantage of using MALDI is that, unlike when using ESI, the signal of sequencing fragments is no longer diluted over multiple charge states, which permits to decrease the overall analysis time from 5 min to 3 min without compromising detection of product ions and associated sequence coverage (Supplementary Tables 12–14).

In summary, the structural design of digital b-PPDE polymers could be finely tuned to fully exploit the unique instrumental configuration of the Synapt G2 for efficient serial sequencing of their constituting blocks. Of note, optimal structural design with appropriate tag distribution is mainly conceived by the writer while the reader can perform de novo sequencing with no prior knowledge of the synthesis history (see Supplementary Protocol). The analytical workflow optimized in this study permits to approach parallelization of sequencing events that can only be achieved in costly instruments such as Fourier transform ion cyclotron resonance (FT-ICR)[21,22], yet in a slower throughput. Beside lowering acquisition time of CID data (typically by a factor of 2 in ESI and a factor of 3 in MALDI for a triblock PPDE), the analytical process is also substantially accelerated and prone to automation. Using ESI, only the *m/z* value of the primary precursor ion needs to be documented whereas, with the previous sequential method, the operator had also to examine MS$^2$ spectra before scheduling pseudo-MS$^3$ experiments with *m/z* values of all secondary precursors. Using MALDI, automation perspectives are even more promising: primary precursor selection is no longer required and automated exploration of different samples on the MALDI plate will be possible in the imaging mode. The number of sequenceable blocks, and so the amount of readable information per chain, is limited by the modest resolving power of our TWIMS cell but the surface sampling capabilities of MALDI opens promising perspectives for a four-dimensional workflow which would allow high throughout reading of large sets of data based on 4D descriptors, namely, spatial coordinates of b-PPDEs on the MALDI plate to be combined with *m/z* ratio, IMS arrival time and CID spectra of their blocks. Storing large amounts of information in multiple triblock oligomers spatially arranged on a surface rather than in long multi-block chains is also potentially advantageous from a synthesis point-of-view.

## Methods
### Materials
Water, methanol (MeOH) and acetonitrile (ACN) used to prepare sample solutions were from SDS (Peypin, France). Formic acid, ammonium acetate, poly-DL-alanine as well as α-cyano-4-hydroxycinnamic acid

(CHCA) used as a MALDI matrix were purchased from Sigma Aldrich (St Louis, MO). All chemicals were used as received without further purification. All b-PPDEs investigated in this study were prepared by automated phosphoramidite chemistry on an Expedite 8900 DNA synthesizer (Applied Biosystems, Waltham, MA), according to synthesis protocols detailed in a recent paper[10].

## Sample preparation

Each b-PPDE sample (1 mg) was dissolved in 1 mL of $H_2O$/ACN (50/50, v/v) supplemented with 0.1% formic acid. For ESI experiments, these stock solutions were further diluted (1/10, v/v) with MeOH containing 3.0 mM ammonium acetate. For MALDI experiments, samples were prepared by mixing, within the same droplet, 1 μL of a saturated solution of CHCA in ACN with 1.5 μL of b-PPDE stock solution. The droplet was then left dried onto the MALDI stainless steel plate.

## Mass spectrometry and ion mobility

High resolution mass spectrometry and traveling wave ion mobility (TWIM) spectrometry experiments were performed with a Synapt G2 HDMS instrument from Waters (Manchester, UK) operated in the negative ion mode. For ESI experiments, the capillary voltage was set to −2.27 kV, the sampling cone voltage to −20 V and the extraction cone voltage to −6 V. The desolvation gas ($N_2$) flow was 100 L.h$^{-1}$ at 35 °C. A syringe pump was used to introduce sample solutions at a 10 μL.min$^{-1}$ flow rate. The MALDI source was equipped with a diode-pumped frequency tripled UV Nd-YAG laser emitting at 355 nm (50 μJ pulses of <3 ns duration) operating at a 1 kHz frequency. In the MS mode, ions were analyzed with the orthogonal acceleration time-of-flight (oa-TOF) mass analyzer which was calibrated with poly-DL-alanine as external standard. In the MS/MS mode, ions were mass selected using a quadrupole and activated in the collision cell. For ESI-MS-(CID)-IMS-MS experiments, precursor ions selected by the quadrupole were activated (26 eV, laboratory frame) in the ion trap, ejected from the trap into a cooling cell (helium flow: 180 mL.min$^{-1}$) at the entrance of the TWIM cell filled with $N_2$ (3.0 mbar). For ESI-MS-(CID)-IMS-(CID)-MS experiments, ions exiting the TWIM cell were activated (34 eV, laboratory frame) in the collision cell and fragments were recorded over the 50–1500 $m/z$ range for 300 s. For MALDI-IMS-(CID)-MS experiments, activation in the collision cell was performed according to the $MS^E$ multiplex fragmentation[20], which consists of alternating the collision energy between 55 eV and 120 eV for 150 scans of 1.00 s with 0.02 s interscan delay; fragments were recorded over the 300–3000 $m/z$ range for 180 s. CID was performed with argon as the collision gas. The TWIMS cell was operated in different conditions of wave height (WH) and wave velocity (WV, in m/s) as a function of ion charge state: for doubly charged species, WH = 30 V and WV = 550 m.s$^{-1}$; for singly charged species, WH = 25 V and WV = 130 m.s$^{-1}$. Instrument control, data acquisition and data processing of all experiments were achieved with the MassLynx 4.1 program provided by Waters. DriftScope 2.1 from Waters was used to prepare 3D plots.

## Data availability

The data that support the findings of this study are available from the corresponding authors upon request.

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

## Acknowledgements

The authors thank the French National Research Agency for financial support (Project shapeNread, grant numbers ANR-19-CE29-0015-01 (L.C.) and ANR-19-CE29-0015-02 (J.-F.L.)). I.S. thanks the Doctorate School *Sciences Chimiques* of Aix Marseille University for her PhD fellowship. L.C. acknowledges support from Spectropole, the Analytical Facility of Aix-Marseille University, by granting a special access to the instruments purchased with European Funding (FEDER OBJ2142-3341).

## Author contributions

L.C. and J.-F.L. conceived and designed the experiments. T.S. and G.O. synthesized all digital polymers while I.S. performed all the experiments and analyzed the data. L.C. wrote the manuscript with contributions from all authors.

## Competing interests

The authors declare no competing interests.
