## [Transparent Peer Review file · Nature Communications]

Acceleration, simplification and potential parallelization of digital polymers sequencing by coupling tandem mass spectrometry with ion mobility

Corresponding Author: Professor Laurence Charles

Version 0:

Reviewer comments:

Reviewer #1

(Remarks to the Author)

This manuscript reports an interesting way to serially sequence the fragments encoding one-byte digital information of a sequence-defined polymer. Utilizing a commercially available mass spectrometer (Waters Synapt G2) and, in particular, ion mobility spectrometry (IMS), the authors demonstrated a sequencing of three one-byte-encoding fragments of a triblock poly(phosphodiester) (PPDE). As an information-storing molecule, sequence-defined polymers inevitably evolve to more complex structures involving a large number of chemically distinct monomers to store a large amount of information in a polymer chain. Tandem mass spectrometry sequencing is currently the only method to decode information stored in a synthetic polymer chain. Therefore, fragmenting a large polymer chain into smaller pieces amenable to tandem mass spectrometry is required. The authors pioneered this idea by implementing a labile chemical bond (NOC bond) to induce fragmentation under conventional ionization conditions for mass spectrometry (Ref. 5). Implementing IMS into the workflow can differentiate each byte-size fragment depending on the relative location within the triblock PPDE. This differentiation also relates to the size of the repeating unit and the total number of larger-sized repeating units within the sequence. The mass tag, originally used for mass differentiation of fragments, also contributes to the time difference of the fragments traveling the IMS cell.

Despite this work's potential, the generality of this technique developed for a specific polymer system remains to be demonstrated. As the authors reported, the separation of the fragments by IMS is limited (practically) to three even though this technique is optimized for the PPDE system. What if the chemical structures of the polymers and their encrypted sequences are way more complex than binary sequences encoded in PPDEs? Also, the MALDI mass can be used for potentially parallel sequencing of individual blocks that are isolable by IMS. However, the parallel sequencing, at least in this paper, has not been demonstrated. Therefore, 'possible parallelization' should be removed from the title.

A few of minor technical comments are below:

1. The predicted tA ranges presented in Figure 3a showed slight differences in the tA trend between inner and final blocks while the 0x1y comonomeric composition is kept constant. In the case of 0810 composition, tA of inner block labelled with tag T8 has a smaller value than that of T9 labelled block whereas the opposite trend is observed in the corresponding final block. Similar difference is also observed in tags T2 and T3. I hope the authors can help me understand why the expected tA trend is not consistent between inner and final blocks.
2. The legend in Figure 3a mentioned "tA = 5.61 ms expected for the first block (B1)..." is confusing as the expected tA value for B1 is 5.58 ms mentioned in the main text. I hope the authors can clarify the sentence.
3. The authors mentioned that to extract "pure" CID spectra, 0.30 ms tA range is sufficient to obtain sequencing fragments of good signal-to-noise ratio. I think the authors can help me understand why 0.30 ms is stated as the minimum tA range to ensure sequencing with citations from other literature.
4. The authors mentioned that the IMS peak width increases with increasing tA, presumably due to diffusion effects in the TWIM cell. I wonder whether the peak width is predictable based on experimental parameters or polymer structure, and if so, whether such prediction could be integrated into the tag selection process to optimize block separation.
5. The authors designed b-PPDEs with blocks defined as octamers. I am curious to know whether smaller block sizes, such as tetramers or hexamers, were considered. Since lower molecular weight blocks may exhibit narrower IMS peaks, wouldn't this approach help reduce peak overlap and allow sequencing of longer polymers?

Reviewer #2

(Remarks to the Author)

The work by Laurence Charles and colleagues introduces a compelling strategy integrating tandem mass spectrometry with ion mobility spectrometry (IMS) to accelerate sequencing of digital polymers, specifically block-truncated poly(phosphodiester)s (b-PPDEs), while probing parallelization potential. By exploiting collision cross section (CCS) differences among blocks, their methodology leverages predictive modeling of arrival time (t_a) variations based on tag structures and 0/1-bit compositions to optimize IMS resolution, enabling parallel sequencing of multiple blocks. Experimental validation through successful decoding of 3-block ("ICR") and 5-block ("PHew!") b-PPDEs demonstrated the feasibility of gas-phase block separation and sequence reconstruction. However, current limitations in IMS resolution restrict practical sequencing to ~3 blocks, though future adoption of cyclic IMS or advanced high-resolution technologies could enhance throughput. Furthermore, preliminary integration of MALDI with IMS-CID workflows enabled surface sampling, establishing a foundation for spatially resolved, high-throughput readouts of encoded information. I think this paper should be accepted, but there are a few minor things that I want them to comment on, and there are a few spelling errors/miscapitalizations that he should correct:

1. I noticed that the tags employed in this work predominantly incorporate benzene ring-containing structures. What is the rationale for adopting such aromatic architectures, and could the tag library be further expanded to include other structural motifs to amplify differences in the arrival time (t_a) values of blocks?
2. Since predicted arrival time (t_a) values are frequently mentioned in the manuscript, could all formulas used to calculate t_a be explicitly listed? This would make the paper's logic clearer and allow readers to directly visualize the computational basis of the predictions.
3. In the manuscript, the abbreviation for " α -cyano-4-hydroxycinnamic acid" is inconsistent. Specifically, HCCA is used in the main text, while CHCA appears in the "Methods" section.

Reviewer #3

(Remarks to the Author)

The manuscript: "Acceleration, simplification and possible parallelization of digital polymers sequencing by coupling tandem mass spectrometry with ion mobility" by Isaure Sergent et al. presents an important advancement in digital polymer sequencing, introducing a novel, bottom-up workflow that integrates tandem mass spectrometry (MS/MS) with ion mobility spectrometry (IMS) and possibly with MALDI. This strategy enables facilitated sequencing of sub-sequences from block-truncated poly(phosphodiester)s in a single analytical run. The approach addresses a limitation of previous methodologies and offers a solution analysis required for reliable and improved sequence reading. The study builds on strong prior work in the field, and the technical achievement is notable. The methodology is sound and carefully executed with sufficient details to be reproduced. However, while the research is timely and high-impact, the current manuscript form is too technical and lacks clarity in several key areas, making it difficult for the broader Nature Communications readership to follow. The novel coupling of IMS with tandem MS has the potential to streamline sequencing workflows and enable automation, which are crucial for scaling digital polymer technologies. However, in its current form, the manuscript is too technical and lacks the clarity expected for a broad audience. A major revision is needed to improve accessibility, provide quantitative benchmarking, and more effectively communicate the conceptual advances. I recommend addressing the points below prior to reconsideration for publication.

1. The authors should clearly articulate the impact and practical advantages of the new method. Specifically, how does it accelerate the sequencing process? A quantitative comparison with the previous MS-based methods is needed. For instance, how much faster can 1 kB of data be sequenced using the proposed approach?
2. Figure 1 is intended to illustrate the comparative workflows but lacks clarity. It should be redrawn with clearly labelled steps, minimized abbreviations, and a visual indication of sample input/output. Consider including a specific example that illustrates sequencing fragments in relation to polymer design. Showing a representative spectral output of these two workflows would be beneficial.
3. The manuscript does not sufficiently explain the TWIMS cell. A brief, clear description of how this component functions within the workflow is necessary for non-expert readers.
4. Scheme 1 should be revised and simplified. Each step of cleavage/separation/analysis should be clearly marked. It is also not obvious where Ti mass tags are introduced or how they relate to the sequencing output. The Figure could be combined with graphical motifs from Supplementary Fig. 1.
5. The principle behind the IMS-based separation should be better described and more explicitly. How exactly do the tags influence separation? Is separation solely dependent on the mass, shape, or charge of fragments, or on tag-specific mobility differences? Which of those factors plays a critical role?
6. For proposed parallelization of the analysis with MALDI, a schematic or representative 3D output plot (position vs. m/z vs. IMS arrival time) should be added to illustrate the power of the method and help readers understand the nature of the sequencing output. Some quantitative data on the benefits derived from this combination of analytical steps should be discussed in the context of readout speed.
7. The authors should address the limitations of the method. How does the proposed workflow scale with increasing polymer molar mass or block number? Are there resolution limits for long sequences or closely related subsequences?
8. It would be interesting to compare directly two MS spectra of the same sequence read by the previous and novel approaches. It should be demonstrated how readable the spectra are and how resolution is changing with the proposed advancements.
9. This work presents a significant step forward in the field of sequence-defined polymers and digital data storage. However, the current manuscript form is too technical and should be revised to make it accessible to non-experts. The technical discussions could be transferred to supporting information. Instead, provide quantitative benchmarking and better

communicate the conceptual advances.

Reviewer #4

(Remarks to the Author)

Summary of key results.

In this manuscript, the authors present an advancement in their previously reported digital polymer (Al Ouahabi et al. *Nat. Commun.* 2017, 8, 967), which encodes information as a sequence of synthetic monomers and decodes it via pseudo MS3 based fragmentation. In both the previous and current work, selective fragmentation at alkoxyamine bonds yields phosphodiester (PDE) blocks in the MS2 spectra, each representing sequences of binary-coded monomers. In previous work, PDE blocks were differentiated by mass tags, which were unique monomers that ensured there was no mass-overlap of blocks with the same binary monomer composition. A separate MS experiment was required to collect and decode the pseudo MS3 spectra for each released PDE block in the polymer. The innovation in this study is the replacement of mass tags with "shape" tags, which allow the PDE block fragments to be separated using ion mobility spectrometry (IMS) and sequenced in a single experiment as they exit the IMS drift tube.

This advantage comes at the expense of the length of the digital polymer. The authors show that not only does the shape tag influence retention time in IMS, but also the composition of binary monomers. They find a linear correlation between the number of 1-bit monomers and IMS arrival time (tA), enabling them to predict the tA of each PDE block and choose a shape tag to ensure adequate separation. This method was used to encode and retrieve the text string "IRC" (3 byte) and "PHew!" (5 byte) using their ASCII binary encoding. However, the 5-byte digital polymer required a severely restricted monomer bit composition to ensure sufficient IMS resolution of each byte-encoding PDE block.

To compensate for this disadvantage, the authors show that their approach is compatible with a MALDI-IMS-(CID)-MS workflow. Using MALDI to read digital polymers could allow for additional information to be stored by the spatial coordinates of the polymer on the MALDI plate.

Significance.

The significance of this work lies in the streamlining of the decoding process by enabling serial sequencing of all PDE blocks in a single MS-(CID)-IMS-(CID)-MS experiment without requiring iterative analysis, extensive sample prep, or time-consuming purification. However, the significance of this advance is more incremental relative to the novelty of their earlier work published in *Nat. Commun.* (Al Ouahabi et al. *Nat. Commun.* 2017, 8, 967) which introduced the molecular design of the digital polymer with controlled fragmentation in MS2 and MS3, allowing for de novo sequencing.

In this work, IMS separation introduces a few limitations to their application in information storage. First, with mass-tags, the authors previously used up to 8 encoding monomers (Laurent et al. *C. R. Chim.* 2021, 24, 69–76), enabling base-8 information storage. In this work, they revert back to binary encoding. Expanding back up to base-8 would require the tA of the PDE blocks to predictably correlate to the overall composition of all 8 monomers, in addition to a range of shape tags which could resolve all PDE blocks. Secondly, again using mass tags instead of shape tags, the authors previously stored and decoded information in polymers of up to 11 PDE blocks (Laurent et al. *C. R. Chim.* 2021, 24, 69–76). With shape tags, practical information storage is limited to 3 PDE blocks. At longer lengths (i.e. 5 blocks) the binary monomer composition is severely restricted, limiting the applicability of longer polymers to universally store information.

Data and methodology.

Analyzing mass spectra fragmentation patterns is outside the scope of my expertise. However, MS fragmentation spectra and tables are nicely annotated in the SI. The authors have previously developed a software to decode MS/MS spectra of digital polymers (Burel et al. *Macromolecules*, 2017, 50, 8290–8296). Were they able to use their software on the spectra collected for this work?

From reading the manuscript, it was not clear if this method could be used for de novo sequencing:

- In the MS-(CID)-IMS-(CID)-MS sequencing routine, it is important in the first MS to choose a precursor ion that is a multiple of the number of PDE blocks in the polymer. Would this be possible with an unknown sample?
- In the introduction, the authors state that MS2 spectra contains masses of individual PDE blocks as well as polymer fragments which enable the reconstruction of the order of the PDE blocks. However, the authors do not show this in the current work in either the main text or the SI. The ability to reconstruct the PDE block sequence from the MS2 spectra is crucial for de novo sequencing, since Figure 3a shows that the order of the arrival time of the PDE blocks during IMS is not inherently preserved.
- Does the digital polymer reader need to know the tA or m/z of the PDE blocks in advance to collect spectra that are clean enough to decode? The authors might also show the crude IMS trace (before each PDE block has been mass-selected) in the SI.

In the MALDI analysis of the digital polymer, all observed fragments must be above 300 m/z, which the authors note does not compromise decoding the PDE sequence. Does this hold true for all monomer compositions? Could the lightest possible

PDE block, with only 0-bit monomers and at the beginning of the polymer, be fully reconstructed with MALDI?

Can the authors estimate how much time is saved by introducing an IMS separation into their sequencing routing? Even small improvements in reading-rate can add up when large amounts of stored information are retrieved.

As the authors report in their work, spatial separation of digital polymers on a DESI plate has already been reported. Digital polymers can additionally be spatially separated using well-plates or even vials. A concrete example showing the unique aspect or benefit of spatial encoding on a MALDI plate would increase the impact of this work.

Version 1:

Reviewer comments:

Reviewer #1

(Remarks to the Author)

Most of the questions raised by the reviewers were addressed in this version of the manuscript. A minor comment is about the so-called "de novo sequencing" of digital polymers by mass spectrometry. Synthetic sequence-defined polymers exhibit unlimited structural complexity arising from the choice of backbone chemistry and monomers. Therefore, there must be some protocols for the chemical structures of digital polymers. If there are protocols, mass sequencing of digital polymers should not be excessively complex. The present sequencing methods are finely tuned for the suggested chemical structures of b-PPDE. My personal opinion is that there will be a consensus for optical chemical structures of digital polymers, and the rest will follow for optimizing decoding processes.

Reviewer #2

(Remarks to the Author)

The authors' responses based on the manuscript of "Acceleration, simplification and possible parallelization of digital polymers sequencing by coupling tandem mass spectrometry with ion mobility" are remarkably consistent and comprehensive, demonstrating a deep understanding of the subject matter. Each answer is well-structured and provides thorough insights, ensuring clarity and accuracy throughout. I am thoroughly satisfied with the quality of the work, which aligns well with the rigorous standards of Nature Communications, making it a strong acceptance.

Reviewer #3

(Remarks to the Author)

The authors have carefully addressed all the points raised in the previous round of review and have improved the manuscript. The revisions have clarified the data presentation and strengthened the impact of the work. In its current form, the manuscript presents a sound methodology, with a detailed description of the experiments that provides sufficient information to ensure reproducibility, robust data analysis, and supports the conclusions. The results are noteworthy, advancing the field of molecular data storage, which addresses important problems arising in modern informational technologies - specifically, how to efficiently store and read data. The work is original, well contextualized within the existing literature, and in my opinion, meets the high standards expected for publication in Nature Communications. I have no further comments, and I recommend acceptance of the manuscript in its current form.

Reviewer #4

(Remarks to the Author)

In my opinion, the authors have satisfactorily addressed the concerns raised by all reviewers. In particular, their addition of a de novo sequencing routine into the SI helped to demonstrate the general utility of their digital polymers. The additional text added to the manuscript's conclusion helped to highlight the advances of this work. This included developing a routine that would allow their digital polymers to be completely decoded in one MS sequence, an important advance in this group's body of work. Their work remains scientifically sound and impactful.

Nonetheless, perhaps because I do not have much expertise in mass spectrometry, I am still not quite convinced that this work reaches the impact required for Nature Communications. The overall design and decoding of their digital polymer remain the same, swapping out mass-tags for shape-tags. And although the author's work developing this sequencing routine for MALDI is exciting and shows a lot of future promise, it does not yet demonstrate the full potential of spatial encoding.

General remarks to the Editor

We first wish to thank all four reviewers for their deep analysis of our work, their suggestions as well as their positive and encouraging comments. In addition to changes made to address their comments and suggestions, the abstract has been slightly modified to comply with the “150 words or fewer” requirement regarding its size. In addition, to best comply with submission guidelines, Scheme 1 was changed to Figure 2 (and following Figures re-numbered) and some figure captions were slightly modified so that they all have a brief title sentence.

Answers to Reviewer #1

This manuscript reports an interesting way to serially sequence the fragments encoding one-byte digital information of a sequence-defined polymer. Utilizing a commercially available mass spectrometer (Waters Synapt G2) and, in particular, ion mobility spectrometry (IMS), the authors demonstrated a sequencing of three one-byte-encoding fragments of a triblock poly(phosphodiester) (PPDE). As an information-storing molecule, sequence-defined polymers inevitably evolve to more complex structures involving a large number of chemically distinct monomers to store a large amount of information in a polymer chain. Tandem mass spectrometry sequencing is currently the only method to decode information stored in a synthetic polymer chain. Therefore, fragmenting a large polymer chain into smaller pieces amenable to tandem mass spectrometry is required. The authors pioneered this idea by implementing a labile chemical bond (NOC bond) to induce fragmentation under conventional ionization conditions for mass spectrometry (Ref. 5). Implementing IMS into the workflow can differentiate each byte-size fragment depending on the relative location within the triblock PPDE. This differentiation also relates to the size of the repeating unit and the total number of larger-sized repeating units within the sequence. The mass tag, originally used for mass differentiation of fragments, also contributes to the time difference of the fragments traveling the IMS cell. Despite this work's potential, the generality of this technique developed for a specific polymer system remains to be demonstrated.

This Reviewer mentions that the reporting coupling has been specifically developed for a polymer system and that its generality remains to be demonstrated. Actually, things have to be considered the other way round: this is the technical configuration of the instrument used here that permits to conceive an accelerated readout method particularly well adapted to the architecture of the investigated polymers. This is clearly mentioned in the last paragraph of the *Introduction*. Of course, this particular method is specific to this polymer system but what remains general is the strategy which consists of designing the structure of digital polymers to take best advantage of techniques available to decode them. This was mentioned as the first concluding remark: “*In summary, the structural design of digital b-PPDE polymers could be finely tuned to fully exploit the unique instrumental configuration of the Synapt G2 for efficient serial sequencing of their constituting blocks.*”

As the authors reported, the separation of the fragments by IMS is limited (practically) to three even though this technique is optimized for the PPDE system. What if the chemical structures of the polymers and their encrypted sequences are way more complex than binary sequences encoded in PPDEs?

We agree that the modest resolving power of our instrument is the main limitation of the proposed approach and that the use of more complex monomer alphabets would further limit the number of separable blocks. Yet, implementing the CID/IMS/CID coupling with a surface sampling technique (here MALDI) permits to compensate the limited IMS resolution by adding another separation dimension. As a result, the same large amounts of information could be stored in (and read from) multiple triblock oligomers arranged in a specific order of the MALDI plate rather than in long multi-

block chains, which can also be advantageous from the synthesis side. This is now best emphasized in the last part of the concluding remarks which has been modified as follows: *“The number of sequenceable blocks, and so the amount of readable information per chain, is limited by the modest resolving power of our TWIMS cell but the surface sampling capabilities of MALDI opens promising perspectives for a four-dimensional workflow which would allow high throughout reading of large sets of data based on 4D descriptors, namely, spatial coordinates of b-PPDEs on the MALDI plate to be combined with m/z ratio, IMS arrival time and CID spectra of their blocks. Storing large amounts of information in multiple triblock oligomers spatially arranged on a surface rather than in long multi-block chains is also potentially advantageous from a synthesis point-of-view.”*

Also, the MALDI mass can be used for potentially parallel sequencing of individual blocks that are isolable by IMS. However, the parallel sequencing, at least in this paper, has not been demonstrated. Therefore, ‘possible parallization’ should be removed from the title.

In the title, the word “possible” has been changed to “potential”.

A few of minor technical comments are below.

1. The predicted tA ranges presented in Figure 3a showed slight differences in the tA trend between inner and final blocks while the 0x1y comonomeric composition is kept constant. In the case of 0810 composition, tA of inner block labelled with tag T8 has a smaller value that of T9 labelled block whereas the opposite trend is observed in the corresponding final block. Similar difference is also observed in tags T2 and T3. I hope the authors can help me understand why the expected tA trend is not consistent between inner and final blocks.

Inner and final blocks do not have the same ω end-group (see Scheme 1) and this structural difference in conjunction with the tag obviously influence the 3D shape of the blocks and so their measured tA. These effects were observed experimentally and we did not perform theoretical calculation and molecular modeling to rationalize them. The following sentence has been added (in the last paragraph on page 5): *“However, the CCS influence of tags was observed to change with the block category but no clear trend could be identified.”*

2. The legend in Figure 3a mentioned “tA = 5.61 ms expected for the first block (B1)...” is confusing as the expected tA value for B1 is 5.58 ms mentioned in the main text. I hope the authors can clarify the sentence.

Sorry for this, this is an error: here, predicted tA if indeed 5.58 ms. This has been corrected in the legend of this Figure (now Figure 4a).

3. The authors mentioned that to extract “pure” CID spectra, 0.30 ms tA range is sufficient to obtain sequencing fragments of good signal-to-noise ratio. I think the authors can help me understand why 0.30 ms is stated as the minimum tA range to ensure sequencing with citations from other literature.

This is not a general rule but an experimental result: the part of IMS peaks (for B2 and B3) that was free of any interference has a width of 0.30 ms and this 0.30 ms range was found to be fine to extract “pure” CID spectra. To clarify this, the sentence *“In particular, although not fully resolved, IMS peaks corresponding to B2 and B3 were large enough to enable extraction of “pure” CID spectra over a 0.30 ms tA range, which was sufficient to obtain sequencing fragments of good signal-to-noise ratio”* has been changed to *“As mentioned before, IMS peaks corresponding to B2 and B3 are not fully resolved but they both exhibit a 0.30 ms tA range free of interference (Supplementary Fig. 6) to enable extraction of “pure” CID spectra, which was sufficient to obtain sequencing fragments of good signal-to-noise ratio (vide infra).”*

4. The authors mentioned that the IMS peak width increases with increasing t_A , presumably due to diffusion effects in the TWIM cell. I wonder whether the peak width is predictable based on experimental parameters or polymer structure, and if so, whether such prediction could be integrated into the tag selection process to optimize block separation.

The width of IMS peaks can also vary in case investigated ions exist as conformers with minor CCS differences, so we did not consider this parameter as a robust factor to be included in the tag selection process.

5. The authors designed b-PPDEs with blocks defined as octamers. I am curious to know whether smaller block sizes, such as tetramers or hexamers, were considered. Since lower molecular weight blocks may exhibit narrower IMS peaks, wouldn't this approach help reduce peak overlap and allow sequencing of longer polymers?

It can indeed be anticipated that blocks of smaller size (and hence lower mass) would exhibit narrower IMS peaks but owing to the quite low peak capacity of the IMS cell, we are not convinced that this will dramatically increase the number of separable blocks whereas this will substantially lower the amount of stored data.

Answers to Reviewer #2

The work by Laurence Charles and colleagues introduces a compelling strategy integrating tandem mass spectrometry with ion mobility spectrometry (IMS) to accelerate sequencing of digital polymers, specifically block-truncated poly(phosphodiester)s (b-PPDEs), while probing parallelization potential. By exploiting collision cross section (CCS) differences among blocks, their methodology leverages predictive modeling of arrival time (t_a) variations based on tag structures and 0/1-bit compositions to optimize IMS resolution, enabling parallel sequencing of multiple blocks. Experimental validation through successful decoding of 3-block ("ICR") and 5-block ("PHew!") b-PPDEs demonstrated the feasibility of gas-phase block separation and sequence reconstruction. However, current limitations in IMS resolution restrict practical sequencing to ~3 blocks, though future adoption of cyclic IMS or advanced high-resolution technologies could enhance throughput. Furthermore, preliminary integration of MALDI with IMS-CID workflows enabled surface sampling, establishing a foundation for spatially resolved, high-throughput readouts of encoded information. I think this paper should be accepted, but there are a few minor things that I want them to comment on, and there are a few spelling errors/miscapitalizations that he should correct:

1. I noticed that the tags employed in this work predominantly incorporate benzene ring-containing structures. What is the rationale for adopting such aromatic architectures, and could the tag library be further expanded to include other structural motifs to amplify differences in the arrival time (t_a) values of blocks?

In a previous study employing small molecular weight model nitroxides (reference 12), it was found that aromatic side-chain substituents give marked IMS responses. Such substituents were therefore an obvious choice for the present proof-of-principle. However, the tag library could indeed be expanded in the future. As recently demonstrated (reference 10), very different types of tags can be synthesized. However, these syntheses require several steps and optimization. Another published study has investigated the possibility to expand even further the tag library (see reference 11): most promising features notably involve multiple aromatic rings but their inclusion as tag in b-PPDEs raises issues that remain to be solved on the synthesis side.

2. Since predicted arrival time (t_a) values are frequently mentioned in the manuscript, could all formulas used to calculate t_a be explicitly listed? This would make the paper's logic clearer and allow readers to directly visualize the computational basis of the predictions.

This is a good suggestion: formulas used for t_A prediction as well as predicted t_A are now reported in Supplementary Table 2, which is referenced in the first paragraph of the section entitled *Predictive distribution of tags in b-PPDEs* on page 6. Subsequent Supplementary Tables have been re-numbered accordingly, in the text and SI.

3. In the manuscript, the abbreviation for " α -cyano-4-hydroxycinnamic acid" is inconsistent. Specifically, HCCA is used in the main text, while CHCA appears in the "Methods" section.

Sorry about this: the same "CHCA" acronym is now used consistently.

Answers to Reviewer #3 (Remarks to the Author):

The manuscript: "Acceleration, simplification and possible parallelization of digital polymer sequencing by coupling tandem mass spectrometry with ion mobility" by Isaure Sergent et al. presents an important advancement in digital polymer sequencing, introducing a novel, bottom-up workflow that integrates tandem mass spectrometry (MS/MS) with ion mobility spectrometry (IMS) and possibly with MALDI. This strategy enables facilitated sequencing of sub-sequences from block-truncated poly(phosphodiester)s in a single analytical run. The approach addresses a limitation of previous methodologies and offers a solution analysis required for reliable and improved sequence reading. The study builds on strong prior work in the field, and the technical achievement is notable. The methodology is sound and carefully executed with sufficient details to be reproduced. However, while the research is timely and high-impact, the current manuscript form is too technical and lacks clarity in several key areas, making it difficult for the broader Nature Communications readership to follow. The novel coupling of IMS with tandem MS has the potential to streamline sequencing workflows and enable automation, which are crucial for scaling digital polymer technologies. However, in its current form, the manuscript is too technical and lacks the clarity expected for a broad audience. A major revision is needed to improve accessibility, provide quantitative benchmarking, and more effectively communicate the conceptual advances. I recommend addressing the points below prior to reconsideration for publication.

By addressing the comments listed below by this Reviewer, we hope that our manuscript will be best suited to the broad audience of Nature Communications.

1. The authors should clearly articulate the impact and practical advantages of the new method. Specifically, how does it accelerate the sequencing process? A quantitative comparison with the previous MS-based methods is needed. For instance, how much faster can 1 kB of data be sequenced using the proposed approach?

This is a very good suggestion: improvements enabled by the proposed coupling compared to the previous sequential method deserve to be better quantified. One first major advantage of the serial approach is acceleration of reading. When using ESI, the acquisition time required to record sequencing fragments is 5 minutes, independently of the number of blocks, whereas it took 3 minutes per block with the previous method. When using MALDI, the sequencing process is even shorter since the lack of multiply charged fragments permits to decrease the acquisition time from 5 to 3 minutes (this was already mentioned at the end of the section entitled *Surface sampling in-line with serial sequencing*). Minimizing the operator's time and automation of the method are other key advantages of the proposed coupling. Accordingly, the following sentences have been added:

- at the bottom of page 6 (for the triblock): "*Most importantly, the acquisition time for this experiment was 5 minutes, that is, nearly two-fold less time than the 9 minutes (3 minutes per block) typically requested for sequencing of triblock b-PPDEs with the previous sequential method.*"

- on page 8 (for the pentablock): *“In addition, acceleration of the reading process is more substantial than in the case of the ‘ICR’ triblock as it took 5 minutes to obtain CID data for all five blocks as compared to 15 minutes when performing sequencing of five blocks in a sequential manner.”*
- on page 13, in the concluding remarks: *“Beside lowering acquisition time of CID data (typically by a factor of 2 in ESI and a factor of 3 in MALDI for a triblock b-PPDE), the analytical process is also substantially accelerated and prone to automation. Using ESI, only the m/z value of the primary precursor ion needs to be documented whereas, with the previous sequential method, the operator had also to examine MS² spectra before scheduling pseudo-MS³ experiments with m/z values of each secondary precursor. Using MALDI, automation perspectives are even more promising: primary precursor selection is no longer required and automated exploration of different samples on the MALDI plate will be possible in the imaging mode.”*

2. Figure 1 is intended to illustrate the comparative workflows but lacks clarity. It should be redrawn with clearly labelled steps, minimized abbreviations, and a visual indication of sample input/output. Consider including a specific example that illustrates sequencing fragments in relation to polymer design. Showing a representative spectral output of these two workflows would be beneficial.

A new version of Figure 1 is now provided and its legend has been completed to improve description of the two workflows as well as the role of tags in each instrumental configuration.

3. The manuscript does not sufficiently explain the TWIMS cell. A brief, clear description of how this component functions within the workflow is necessary for non-expert readers.

Explanation of how the TWIMS cell works can be found in dedicated literature and we are not convinced that too many details on how this device actually functions will help much non-expert readers. Yet, the general definition of IMS given very early in the two first sentences of the Results & discussion section has been completed by a brief description of the TWIMS technology (on page 4) and an additional reference: *“The TWIMS technology used in Synapt instruments employs a sequence of potential waves propagating through the mobility cell to propel ions through the buffer gas.⁹ Ions with larger CCS experiencing more collisions with the gas have their mobility reduced below the velocity of the wave and roll over the potential wave crest more often than more mobile ions of lower CCS that hence travel faster through the cell. Ion separation is typically optimized by varying the wave velocity (WV) and the wave height (WH).”*

4. Scheme 1 should be revised and simplified. Each step of cleavage/separation/analysis should be clearly marked. It is also not obvious where Ti mass tags are introduced or how they relate to the sequencing output. The Figure could be combined with graphical motifs from Supplementary Fig. 1.

The goal of Scheme 1 is not to describe each step of cleavage/separation/analysis but to detail the structure of each block according to its category, which relates to its initial location in the intact chain shown in panel a. However, we agree that in the displayed structure, Ti labels used for tags can be made clearer, which has been done in the new version of Scheme 1 (now Figure 4).

5. The principle behind the IMS-based separation should be better described and more explicitly. How exactly do the tags influence separation? Is separation solely dependent on the mass, shape, or charge of fragments, or on tag-specific mobility differences? Which of those factors plays a critical role?

More details on the principle behind the IMS-based separation have been given while addressing comment #3 of this Reviewer.

The section entitled *IMS behavior of PDE blocks* is all about factors influencing block mobility, which was summarized at the end of the first paragraph: “*t_A measured for [Bi – 2H]²⁻ ions in IMS are observed to depend on the block category (first, inner or last), as well as on their 0/1 comonomeric composition and the structure of their tag but, interestingly, not on their sequence.*” The respective influence of these factors cannot be clearly quantified: this is the reason why we have constructed a predictive approach based on the linear trend of t_A as a function of the number of 1-bits (Fig. 2).

6. For proposed parallelization of the analysis with MALDI, a schematic or representative 3D output plot (position vs. m/z vs. IMS arrival time) should be added to illustrate the power of the method and help readers understand the nature of the sequencing output. Some quantitative data on the benefits derived from this combination of analytical steps should be discussed in the context of readout speed.

We do not believe that the suggested representative 3D output will be helpful: the proposed parallelization of the analysis with MALDI typically relies on the well-known imaging concept, as clearly mentioned from the beginning of the section entitled *Surface sampling in-line with serial sequencing*.

Quantitative data on the benefits derived from this combination of analytical steps have been provided while addressing comment #1 from this Reviewer.

7. The authors should address the limitations of the method. How does the proposed workflow scale with increasing polymer molar mass or block number? Are there resolution limits for long sequences or closely related subsequences?

We believe we have largely addressed the main limitation of the method, which is the low peak capacity of the IMS cell. As clearly written in the text, practical use of this analytical workflow is limited to reading PPDE triblocks: this is the reason why we proposed to switch from ESI to MALDI in order to take advantage of surface as an additional storage dimension.

8. It would be interesting to compare directly two MS spectra of the same sequence read by the previous and novel approaches. It should be demonstrated how readable the spectra are and how resolution is changing with the proposed advancements.

CID spectra used for block sequencing are all measured with the same TOF mass analyzer, regardless of the analytical workflow (sequential vs serial): as a result, mass resolution is not changing. The only difference between the two analytical workflows is the acquisition time to record CID spectra: 3 minutes for sequential sequencing vs 5 minutes for serial sequencing in order to compensate for signal attenuation due to the IMS step. This is now mentioned in the revised version of the manuscript, as a result of addressing comment #1 from this Reviewer.

9. This work presents a significant step forward in the field of sequence-defined polymers and digital data storage. However, the current manuscript form is too technical and should be revised to make it accessible to non-experts. The technical discussions could be transferred to supporting information. Instead, provide quantitative benchmarking and better communicate the conceptual advances.

This last points seems to be a summary of previous comments that were hopefully addressed properly.

Answers to Reviewer #4

Summary of key results – In this manuscript, the authors present an advancement in their previously reported digital polymer (Al Ouahabi et al. Nat. Commun. 2017, 8, 967), which encodes information as a sequence of synthetic monomers and decodes it via pseudo MS3 based fragmentation. In both the previous and current work, selective fragmentation at alkoxyamine bonds yields phosphodiester (PDE) blocks in the MS2 spectra, each representing sequences of binary-coded monomers. In previous work, PDE blocks were differentiated by mass tags, which were unique monomers that ensured there was no mass-overlap of blocks with the same binary monomer composition. A separate MS experiment was required to collect and decode the pseudo MS3 spectra for each released PDE block in the polymer. The innovation in this study is the replacement of mass tags with “shape” tags, which allow the PDE block fragments to be separated using ion mobility spectrometry (IMS) and sequenced in a single experiment as they exit the IMS drift tube. This advantage comes at the expense of the length of the digital polymer. The authors show that not only does the shape tag influence retention time in IMS, but also the composition of binary monomers. They find a linear correlation between the number of 1-bit monomers and IMS arrival time (tA), enabling them to predict the tA of each PDE block and choose a shape tag to ensure adequate separation. This method was used to encode and retrieve the text string “IRC” (3 byte) and “PHew!” (5 byte) using their ASCII binary encoding. However, the 5-byte digital polymer required a severely restricted monomer bit composition to ensure sufficient IMS resolution of each byte-encoding PDE block. To compensate for this disadvantage, the authors show that their approach is compatible with a MALDI-IMS-(CID)-MS workflow. Using MALDI to read digital polymers could allow for additional information to be stored by the spatial coordinates of the polymer on the MALDI plate.

This is a perfect summary of our study, showing that this Reviewer has fully understood all the details and issues of this study.

Significance – The significance of this work lies in the streamlining of the decoding process by enabling serial sequencing of all PDE blocks in a single MS-(CID)-IMS-(CID)-MS experiment without requiring iterative analysis, extensive sample prep, or time-consuming purification. However, the significance of this advance is more incremental relative to the novelty of their earlier work published in Nat. Commun. (Al Ouahabi et al. Nat. Commun. 2017, 8, 967) which introduced the molecular design of the digital polymer with controlled fragmentation in MS2 and MS3, allowing for de novo sequencing. In this work, IMS separation introduces a few limitations to their application in information storage. First, with mass-tags, the authors previously used up to 8 encoding monomers (Laurent et al. C. R. Chim. 2021, 24, 69–76), enabling base-8 information storage. In this work, they revert back to binary encoding. Expanding back up to base-8 would require the tA of the PDE blocks to predictably correlate to the overall composition of all 8 monomers, in addition to a range of shape tags which could resolve all PDE blocks. Secondly, again using mass tags instead of shape tags, the authors previously stored and decoded information in polymers of up to 11 PDE blocks (Laurent et al. C. R. Chim. 2021, 24, 69–76). With shape tags, practical information storage is limited to 3 PDE blocks. At longer lengths (i.e. 5 blocks) the binary monomer composition is severely restricted, limiting the applicability of longer polymers to universally store information.

This Reviewer’s analysis of our past works is correct but we do not consider the outputs of the present study as incremental advances. We agree that the peak capacity of our IMS cell is disappointing, which led us to exploit an additional storage dimension (surfaces) by using a surface sampling technique (MALDI) to compensate for the limited IMS resolution. Performance of the MALDI approach have been validated and notably offer highly promising perspectives based on two major advantages. First, when developed in the MALDI imaging mode, the analytical workflow will be fully automated and switching from one sample to another much faster than with ESI. Second, preparing short triblock chains as requested to ensure IMS separation of all blocks is not necessary a drawback from the synthesis side since both sample purity and size monodispersity are easier to achieve for short chains. These two features have been added in the text while addressing comment #2 of Reviewer 1 and comment #1 of Reviewer 3.

Data and methodology – Analyzing mass spectra fragmentation patterns is outside the scope of my expertise. However, MS fragmentation spectra and tables are nicely annotated in the SI. The authors have previously developed a software to decode MS/MS spectra of digital polymers (Burel et al. *Macromolecules*, 2017, 50, 8290–8296). Were they able to use their software on the spectra collected for this work?

We did not use our MS-DECODER software in the present study. Since the MALDI imaging approach (still to be developed) is much more promising than serial sequencing with ESI, we chose to focus our efforts on implementing the MALDI coupling in MS-DECODER so that the software is capable to associate sample coordinates, IMS arrival times and CID data in an automated manner.

From reading the manuscript, it was not clear if this method could be used for de novo sequencing:

• In the MS-(CID)-IMS-(CID)-MS sequencing routine, it is important in the first MS to choose a precursor ion that is a multiple of the number of PDE blocks in the polymer. Would this be possible with an unknown sample?

• In the introduction, the authors state that MS2 spectra contains masses of individual PDE blocks as well as polymer fragments which enable the reconstruction of the order of the PDE blocks. However, the authors do not show this in the current work in either the main text or the SI. The ability to reconstruct the PDE block sequence from the MS2 spectra is crucial for de novo sequencing, since Figure 3a shows that the order of the arrival time of the PDE blocks during IMS is not inherently preserved.

• Does the digital polymer reader need to know the tA or m/z of the PDE blocks in advance to collect spectra that are clean enough to decode? The authors might also show the crude IMS trace (before each PDE block has been mass-selected) in the SI.

This is an excellent point that indeed deserves to be addressed. Yes, the method can be used for de novo sequencing: this is now mentioned in the concluding remarks as “*Of note, optimal structural design with appropriate tag distribution is mainly conceived by the writer while the reader can perform de novo sequencing with no prior knowledge of the synthesis history (see Supplementary Protocol)*”. This protocol placed in the SI addresses all questions raised by this Reviewer.

In the MALDI analysis of the digital polymer, all observed fragments must be above 300 m/z, which the authors note does not compromise decoding the PDE sequence. Does this hold true for all monomer compositions? Could the lightest possible PDE block, with only 0-bit monomers and at the beginning of the polymer, be fully reconstructed with MALDI?

Yes, this statement holds true for all monomer compositions thanks to i) the complementary of fragments series and ii) production of (nearly) complete fragment series: information contained in lacking small fragments in MALDI-CID spectra can be found from the analysis of large product ions of the complementary series, as can be seen from Supplementary Tables 12-14. Even the lightest possible PDE block composed of 0-bits only can be safely reconstructed (although it is not really necessary to sequence such a block as its 0_81_0 composition can be readily identified from its mass). The mass of 0-bit is 138 Da so for B1 block with 00000000 sequence, missing fragments would be d1, d2 and c2 that are used to identify the first and second units (this information can be obtained from m/z differences between z8 and z7 and between z7 and z6 for example) as well as z1 and y1 that are used to identify the last unit (this information can also be obtained from w1 or x1).

Can the authors estimate how much time is saved by introducing an IMS separation into their sequencing routing? Even small improvements in reading-rate can add up when large amounts of stored information are retrieved.

The same question was raised by Reviewer 3 (comment #1) and is now addressed in the revised version. Briefly, as compared to sequential sequencing which requires 3 min acquisition per block for CID data, serial sequencing takes 5 min (with ESI) and 3 min (with MALDI) to record CID data of all blocks, regardless of the number of blocks.

As the authors report in their work, spatial separation of digital polymers on a DESI plate has already been reported. Digital polymers can additionally be spatially separated using well-plates or even vials. A concrete example showing the unique aspect or benefit of spatial encoding on a MALDI plate would increase the impact of this work.

The CID/IMS coupling in the MALDI imaging mode is currently in development and is somehow out of the scope of the present study. Work is still needed to answer numerous technical questions such as matrix deposition (could digital polymers be stored on MALDI plates together with the matrix for long periods of time or should the matrix be deposited just before running analysis?), spatial resolution (what are the best techniques for deposition of polymers with optimal spatial storage density? for deposition of the matrix to avoid polymer delocalization?), and software developments to implement outputs of the coupling in MS-DECODER.

Replies to Reviewers' comments – NCOMMS-2526576A

Comment from Reviewer #1

Most of the questions raised by the reviewers were addressed in this version of the manuscript. A minor comment is about the so-called "de novo sequencing" of digital polymers by mass spectrometry. Synthetic sequence-defined polymers exhibit unlimited structural complexity arising from the choice of backbone chemistry and monomers. Therefore, there must be some protocols for the chemical structures of digital polymers. If there are protocols, mass sequencing of digital polymers should not be excessively complex. The present sequencing methods are finely tuned for the suggested chemical structures of b-PPDE. My personal opinion is that there will be a consensus for optical chemical structures of digital polymers, and the rest will follow for optimizing decoding processes.

We wish to thank this Reviewer for the supportive comments.

Comment from Reviewer #2

None.

Comment from Reviewer #3

The authors have carefully addressed all the points raised in the previous round of review and have improved the manuscript. The revisions have clarified the data presentation and strengthened the impact of the work. In its current form, the manuscript presents a sound methodology, with a detailed description of the experiments that provides sufficient information to ensure reproducibility, robust data analysis, and supports the conclusions. The results are noteworthy, advancing the field of molecular data storage, which addresses important problems arising in modern informational technologies - specifically, how to efficiently store and read data. The work is original, well contextualized within the existing literature, and in my opinion, meets the high standards expected for publication in Nature Communications. I have no further comments, and I recommend acceptance of the manuscript in its current form.

We wish to thank this Reviewer for the supportive comments.

Comment from Reviewer #4

In my opinion, the authors have satisfactorily addressed the concerns raised by all reviewers. In particular, their addition of a de novo sequencing routine into the SI helped to demonstrate the general utility of their digital polymers. The additional text added to the manuscript's conclusion helped to highlight the advances of this work. This included developing a routine that would allow their digital polymers to be completely decoded in one MS sequence, an important advance in this group's body of work. Their work remains scientifically sound and impactful. Nonetheless, perhaps because I do not have much expertise in mass spectrometry, I am still not quite convinced that this work reaches the impact required for Nature Communications. The overall design and decoding of their digital polymer remain the same, swapping out mass-tags for shape-tags. And although the author's work developing this sequencing routine for MALDI is exciting and shows a lot of future promise, it does not yet demonstrate the full potential of spatial encoding.

We wish to thank this Reviewer for the supportive comments and enthusiasm regarding promise of the MALDI version of the method. Regarding novelty, we feel that this Reviewer indeed underestimates the impact of our study, on both the analytical and structural design side. While performing CID before or after IMS has become routine, this is the first report of performing CID before AND after IMS. Of course, the architecture of our digital polymers is perfectly suited for such an analytical configuration but "swapping out mass-tags for shape-tags" is not so straightforward.